# Air Pollution and Preterm Birth: Do Air Pollution Changes over Time Influence Risk in Consecutive Pregnancies among Low-Risk Women?

**DOI:** 10.3390/ijerph16183365

**Published:** 2019-09-12

**Authors:** Pauline Mendola, Carrie Nobles, Andrew Williams, Seth Sherman, Jenna Kanner, Indulaxmi Seeni, Katherine Grantz

**Affiliations:** 1Division of Intramural Population Health Research, Epidemiology Branch, *Eunice Kennedy Shriver* National Institute of Child Health and Human Development, 6710B Rockledge Drive, Bethesda, MD 20895, USA; Carrie.nobles@nih.gov (C.N.); andrew.d.williams@aol.com (A.W.); m.kanner@nih.gov (J.K.); katherine.grantz@nih.gov (K.G.); 2The Emmes Company, 401 N Washington St # 700, Rockville, MD 20850, USA; ssherman@emmes.com; 3University of California Davis School of Medicine, 4610 X Street, Sacramento, CA 95817, USA; icseeni@gmail.com

**Keywords:** air pollution, preterm birth, pregnancy, sulfur dioxide, ozone, nitrogen oxides, carbon monoxide, particulate matter

## Abstract

Since the 2000s, air pollution has generally continued to decrease in the U.S. To investigate preterm birth (PTB) risk associated with air pollutants in two consecutive pregnancies, we estimated exposures using modified Community Multiscale Air Quality models linked to the NICHD Consecutive Pregnancy Study. Electronic medical records for delivery admissions were available for 50,005 women with singleton births in 20 Utah-based hospitals between 2002–2010. We categorized whole pregnancy average exposures as high (>75th percentile), moderate (25–75) and low (<25). Modified Poisson regression estimated second pregnancy PTB risk associated with persistent high and moderate exposure, and increasing or decreasing exposure, compared to persistent low exposure. Analyses were adjusted for prior PTB, interpregnancy interval and demographic and clinical characteristics. Second pregnancy PTB risk was increased when exposure stayed high for sulfur dioxide (32%), ozone (17%), nitrogen oxides (24%), nitrogen dioxide (43%), carbon monoxide (31%) and for particles < 10 microns (29%) versus consistently low exposure. PTB risk tended to increase to a lesser extent for repeated PTB (19–21%) than for women without a prior PTB (22–79%) when exposure increased or stayed high. Area-level changes in air pollution exposure appear to have important consequences in consecutive pregnancies with increasing exposure associated with higher risk.

## 1. Introduction

Preterm birth has been associated with air pollution exposure, but no prior studies have examined the change in risk over time in an existing cohort. The NICHD Consecutive Pregnancy Study (2002–2010) contains detailed clinical data on deliveries for more than 50,000 women with at least two pregnancies during the study period [1]. On average, criteria air pollutants in the United States (U.S.) decreased over the 2000’s [2] while the preterm birth rate peaked in 2006 at 12.8% and declined to 11.9% in 2010 [3]. During this period of dynamic change, we evaluated the air-pollution associated preterm birth risk in this cohort of low-risk women.

Considering that continued progress to improve air quality is challenged by the global impact of extreme weather events and climate change, which may increase pollution after decades of improvements, it is important to characterize the link between changes in exposure and population health to inform policy [4]. Even if anthropogenic air pollution stays constant, increases in ambient temperature will be associated with increased ozone (O_3_) and precursors of particulate matter < 2.5 microns (PM_2.5_), as well as a higher incidence of wildfire exposures [5]. A recent meta-analysis reported whole pregnancy exposure to O_3_, carbon monoxide (CO) and particulate matter significantly increased preterm risk [6]. Preterm birth is an important population health indicator with a global incidence of 15 million births each year [7]. It is a leading cause of infant mortality and is associated with long-term adverse effects on neurodevelopment, cardiovascular, renal, metabolic and pulmonary health [8]. The rate of preterm birth is higher in the U.S. than in other developed countries, but changes to ambient environmental exposures, such as the retirement of power plants using fossil fuels [9], can further lower risk and mitigate disparities. Following women over time to determine if their preterm birth risks are potentially influenced by changing ambient air pollution can provide evidence on the potential effectiveness of pollution control measures to improve health outcomes.

In addition to the potential for changes in risk associated with the ambient environment, changes in preterm birth risk over time are also influenced by women’s reproductive history. A recent meta-analysis reported an average absolute recurrent spontaneous preterm birth rate of 30% [10] and we observed a similar recurrence risk in the NICHD Consecutive Pregnancy Study data [1].

Our aim was to examine the risk of preterm birth in a second pregnancy based on the change in air pollutant exposure since the first birth in a cohort of low-risk U.S. women. This is important to demonstrate the human health impacts of improved or degraded air quality over time on preterm birth, a key indicator of population health. We hypothesized that improving air quality would be associated with a lower risk of subsequent preterm birth, and that increased preterm birth risk would be observed when air pollution stayed high over time. We also evaluated potential effect modification by prior preterm birth and hypothesized that women without a prior history, generally at lower risk, would have a higher preterm birth risk associated with air pollution exposure.

## 2. Materials and Methods

The NICHD Consecutive Pregnancy Study [1] is based on electronic medical records for hospital delivery admissions among 50,005 women with singleton births in 20 Utah-based hospitals from the same medical care system between 2002–2010. Institutional Review Board approval was obtained at all participating institutions, and the final dataset was anonymized. The first two consecutive pregnancies observed comprised the analytic dataset. Approximately 20% of women (n = 10,325) delivered at two different study hospitals, with 30% of those (n = 3112) associated with a hospital closure. All outcome and covariate information were derived from the medical records and discharge summaries. Preterm birth was defined as the best clinical estimate of gestational age < 37 completed weeks.

Air pollution exposure was estimated for each pregnancy using modified Community Multi-Scale Air Quality models and averaged over the delivery hospital referral region fused with existing monitor data using inverse distance weighting [1]. Exposure models for sulfur dioxide (SO_2_), O_3_, nitrogen oxides (NO_x_), nitrogen dioxide (NO_2_), CO, particulate matter < 10 microns (PM_10_), and particulate matter < 2.5 microns (PM_2.5_) were also weighted to reflect the population density in order to discount areas where women were unlikely to live or work.

Based on the exposure levels in the first observed pregnancy (Appendix A), we categorized whole pregnancy exposures as high (>75th percentile), moderate (25th to 75th percentile) and low (<25th percentile). Exposures were also calculated truncated at 28 gestational weeks (Appendix A) in order to consider the shorter gestational length for preterm births. Second pregnancies were evaluated based on first pregnancy cut points and characterized based on the change from the first pregnancy as stayed high (high in both pregnancies), stayed moderate (moderate in both pregnancies), increasing (moved from low or moderate to moderate or high) or decreasing (moved from high or moderate to moderate or low). Exposures that stayed low were the reference category.

Relative risks and 95% confidence intervals were estimated using single-pollutant, modified Poisson regression models with robust errors [11] to correct for the over-estimation of variance in a traditional Poisson model. We estimated second pregnancy preterm birth risk associated with whole pregnancy average persistent high and moderate exposure, and increasing or decreasing exposure, compared to persistent low exposure. Missing data were addressed using multiple imputations, using chained equations to generate ten datasets [12]. Analyses were run independently in each imputed dataset with the effect estimates and variability pooled to provide a summary estimate. Analyses were adjusted for prior preterm birth (yes/no in the first observed pregnancy), interpregnancy interval (years), and covariates recorded in the second pregnancy: maternal age (years), pre-pregnancy body mass index (kg/m^3^), race/ethnicity (Non-Hispanic white, Non-Hispanic black, Hispanic, Asian, other/unknown), marital status (married, unmarried), insurance status (public, private), parity (1, 2+), smoking and alcohol use during pregnancy (yes, no, unknown) and history of maternal asthma (yes/no).

Given the strength of prior preterm birth as a predictor of preterm birth in the second pregnancy, we repeated the main analyses evaluating interaction by prior preterm birth status. To assess the robustness of our findings, we also repeated the whole pregnancy average analyses using exposures truncated at 28 gestational weeks and examined the subset of consecutive pregnancies among nulliparas at study onset. To further explore the potential impact of changes over time on second pregnancy preterm birth risk, we evaluated year of conception for the second pregnancy, and the change in season of conception between the first and second pregnancy (stayed warm—reference, warm to cold, cold to warm and stayed cold).

All analyses were conducted using SAS 9.4 (SAS Institute, Cary, NC, USA).

## 3. Results

Infants were delivered preterm in 7.6% of both first and second observed pregnancies (Table 1). The women were generally low risk, with the majority Non-Hispanic white, married women with private insurance. Second pregnancy preterm births were more common among older mothers, women with slightly shorter inter-pregnancy intervals, non-white women, those who were unmarried and had public insurance, as well as mothers who were multiparous and reported smoking and alcohol use during pregnancy. As anticipated, having a prior preterm birth was associated with a higher proportion of preterm birth in the second pregnancy (30.6%) compared with a prior term birth (5.7%) and mothers with asthma were more likely to have a preterm birth.

Exposure to air pollutants were characterized as high, moderate and low based on the percentiles for whole pregnancy exposure in the first observed pregnancy (Appendix A). Pollutants levels tended to decrease between the first and second observed pregnancy, but the changes were not consistent in magnitude (Appendix A) with less decrease over time observed for SO_2_, PM_10_ and PM_2.5_. Overall, women were more likely to experience a decrease in pollutant exposure with approximately 7–12% of women having a category shift to higher exposure in the second pregnancy compared to 27–59% shifting to a lower exposure category (Appendix A). PM_2.5_ was an exception, with a similar proportion of women, approximately 30%, experiencing an increase and a decrease in exposure.

When exposures stayed high in both the first and second pregnancy, compared with persistent low exposure, the relative risk of preterm birth in the second pregnancy was significantly elevated for all pollutants except for PM_2.5_ (Table 2). The highest risks were observed for NO_2_ (43% increase) and SO_2_ (32% increase). When exposure stayed moderate, we observed an increased risk associated with SO_2_ (12% increase) and a decreased risk for PM_2.5_ exposure (18% decrease). Similar to the results for persistent high exposure, increasing exposure over time was generally associated with higher PTB risk, although the results for NO_x_ were not significant. Decreasing exposure over time resulted in lower risk estimates, although still significantly elevated for SO_2_, NO_2_ and PM_10_. For example, the relative risk for SO_2_ when exposure increased was 1.41 versus 1.17 when exposure decreased over time.

Assessing the interaction by prior preterm birth status (Table 3), the risks for persistent high exposure and for increasing exposure over time often appear to be associated with higher risk for women without a prior history of preterm birth, with significant interactions based on preterm birth history for NO_2_, SO_2_, O_3_ and CO. When exposure stayed high, SO_2_ risk was fairly consistent between women with and without a prior history (21% vs. 37% increase, *p* interaction = 0.33), whereas greater risk was observed for NO_2_ among women without a prior preterm birth compared to women with a prior preterm birth (56% vs. 18% increase, *p* interaction = 0.025). Increasing exposure suggested similar risks by prior preterm birth status for PM_10_ (21% vs. 22%, *p* interaction = 0.90), but significantly higher risk for women without a prior preterm birth associated with increasing NO_2_, SO_2_, O_3_ and CO compared to women with a prior preterm birth history.

Restricting the exposure windows to the first 28 weeks of gestation yielded a similar pattern of results, but the effects were somewhat attenuated (Appendix A). Significant findings remained for SO_2_, NO_2_ and PM_10_. When considering only women who were nulliparous at the beginning of the study (n = 27,137, 54.3%), change in exposure over time was associated with a similar pattern of results as the main analysis, although increasing exposure was associated with somewhat higher point estimates in this group (Appendix A). For example, increasing ozone was associated with an 80% increase in risk among nulliparas compared to a 48% increase in the full sample. Examining risk by prior preterm birth among nulliparous women at study entry (Appendix A), O_3_ was associated with increased risk when exposure stayed high regardless of prior preterm birth status, whereas increasing exposure was significantly elevated (*p* for interaction > 0.05) among women without a prior preterm birth for NO_2_, CO and PM_2.5_. Adding year of conception for the second pregnancy or an indicator variable to measure the change in the season of conception from the first to second pregnancy into the models did not substantively change any of the findings (data not shown).

## 4. Discussion

In a cohort of low-risk U.S. women, we found that preterm birth risks were significantly increased in association with persistent high exposure to all criteria air pollutants except for PM_2.5_. Increasing exposure over time was associated with higher preterm birth risk, while decreasing exposure was associated with lower risk estimates. Women without a prior preterm birth experienced a higher risk for an incident preterm birth in the second pregnancy associated with pollution.

During the study period, our cohort had a preterm birth rate of 7.6% in comparison to approximately 12% for the U.S. overall. This low-risk group experienced moderate levels of chronic air pollution exposure and exposure decreased over time for a substantial proportion of the women, ranging from 27% for SO_2_ to 59% for O_3_. Increasing exposure was less common, experienced by only 3% of women with respect to O_3_, but 30–32% of women had increased PM_2.5_ exposure in the second observed pregnancy compared to their first. A prior history of preterm birth is a major risk factor for preterm birth in a subsequent pregnancy [10] and, in our data, this was clearly the case. We also observed that women without a prior history generally had higher preterm birth risks associated with air pollution compared to women with a prior preterm birth. Among higher risk women, the marginal increases associated with air pollution were often not statistically significant, probably due to the competing risks of their history and factors associated with that history. In most exposure scenarios, nulliparous women at study entry also had higher preterm birth risk estimates associated with air pollutants. Taken together, these findings support the notion that the strongest preterm birth risks associated with higher air pollution exposure and increasing exposure over time were observed among the lowest risk women, those without a prior history of preterm birth and nulliparas.

In contrast to many studies of preterm birth [6,13,14], we found no increased risk associated with PM_2.5_ and many of the point estimates we observed were below 1.00. It may be that our chronic average categorization is not an ideal fit for this exposure, given that the Utah Valley experiences inversions and high PM exposures during peak events in the winter. We also noted that nearly one-third of women experienced an increased average PM_2.5_ exposure between the first and second pregnancy, which may be due to wildfire exposures. A more detailed assessment of short-term PM_2.5_ exposures is warranted to consider those factors.

This study takes advantage of a nearly decade-long observational cohort with detailed clinical information on women with more than one delivery in a defined geographic area. The women were low-risk, but still experienced significant increases in preterm birth related to moderate levels of criteria air pollutants. Given the rich clinical data available, we were able to adjust for maternal demographic characteristics and factors, such as interpregnancy interval [15], prior preterm birth [10] and maternal asthma [16,17], all of which are known to be associated with preterm risk, but may not be available in administrative data. We are limited in our exposure assessment because we do not have information on maternal address and assume that the women live in the area covered by their delivery hospital. Even among women who delivered in two different hospitals over the course of the study period, we assume each delivery hospital referral region estimates their local area exposure for that pregnancy. While this will introduce some measurement error, it will likely be non-differential with respect to subsequent preterm delivery and a more broad local exposure area will account for some local mobility as women move around their neighborhoods to work, school and other activities. We also acknowledge that changes in meteorological factors, such as ambient temperature, have been associated with preterm birth risk independent of air pollution [18] and those exposures merit further research.

Our categorical chronic exposure scenario was chosen to examine the bigger picture changes over time that could reasonably be observed, and that might be associated with policy-relevant changes in preterm birth risk associated with these dynamic exposures. Preterm birth risks associated with changing exposure were not influenced by changes to the season of conception between the first and second birth or adjustment for the year of conception of the second pregnancy.

## 5. Conclusions

In this study of low-risk women with consecutive pregnancies in a large cohort, we observed increased risk over time for nearly all criteria pollutants when exposure remained high or increased over time. Decreases in exposure over time were associated with comparatively lower risk estimates. Women without a prior preterm birth experienced a higher risk for an incident preterm birth in the second pregnancy associated with pollution.

These findings suggest that improvements in air quality could reduce related preterm birth risks, but failure to address persistent air pollution results in higher risk, particularly for nulliparous women and those without a prior preterm birth history.

## Figures and Tables

**Table 1 ijerph-16-03365-t001:** Characteristics of the study sample by preterm birth status in the second pregnancy among 50,005 mothers, NICHD Consecutive Pregnancy Study, 2002–2010.

	Preterm Birth N = 3806 (7.6%)		No Preterm Birth N = 46,199 (92.4%)		*p*
	Mean	SD	Mean	SD	
Age (years)	28.1	4.6	27.8	5.1	<0.001
BMI (kg/m^2^)	25.1	5.7	25.2	6.2	0.32
Interpregnancy interval (years)	2.39	1.11	2.50	1.01	<0.001
	N	%	N	%	
Race/ethnicity					<0.001
Non-Hispanic white	3210	84.3	39,905	86.4	
Non-Hispanic black	25	0.7	190	0.4	
Hispanic	438	11.5	4934	10.7	
Asian	96	2.5	909	2	
Other/unknown	37	1	261	0.6	
Marital status					<0.001
Married	3223	94.7	41,869	90.6	
Not married	583	15.3	4330	9.4	
Insurance					<0.001
Public	1223	32.1	11,938	25.8	
Private	2583	67.9	34,261	74.2	
Parity					<0.001
1	1917	50.4	25,210	54.6	
2+	1889	49.6	20,989	45.4	
Smoking					<0.001
Yes	285	7.5	1300	2.8	
No	3517	92.4	44,856	97.1	
Unknown	4	0.1	43	0.1	
Alcohol use					<0.001
Yes	124	3.3	649	1.4	
No	2671	96.5	45,428	98.3	
Unknown	11	0.3	122	0.3	
PTB in prior pregnancy					<0.001
Yes	1163	30.6	2623	5.7	
No	2643	69.4	43,576	94.3	
Maternal asthma					<0.001
Yes	325	8.5	2844	6.2	
No	3481	91.5	43,355	93.8	

**Table 2 ijerph-16-03365-t002:** The adjusted relative risk of preterm birth in second pregnancy by change in whole pregnancy average criteria pollutant exposure levels from first to second pregnancy among 50,005 mothers, NICHD Consecutive Pregnancy Study, 2002–2010.

	Second Pregnancy Preterm Birth			
Change in Whole Pregnancy Exposure from First to Second Observed Pregnancy	RR	LCL	UCL	*p*
Stay high vs. stay low				
SO_2_	1.32	1.17	1.49	<0.0001 *
O_3_	1.17	1.00	1.37	0.046 *
NO_x_	1.24	1.09	1.41	0.001 *
NO_2_	1.43	1.27	1.61	<0.0001 *
CO	1.31	1.15	1.49	<0.0001 *
PM_2.5_	1.05	0.90	1.22	0.53
PM_10_	1.29	1.14	1.45	<0.0001 *
Stay moderate vs. stay low				
SO_2_	1.12	1.01	1.24	0.024
O_3_	0.97	0.84	1.12	0.67
NO_x_	0.89	0.78	1.00	0.053
NO_2_	0.93	0.83	1.05	0.25
CO	0.89	0.79	1.00	0.06
PM_2.5_	0.82	0.73	0.93	0.002 *
PM_10_	1.03	0.94	1.14	0.49
Increase vs. stay low				
SO_2_	1.41	1.27	1.56	<0.0001 *
O_3_	1.48	1.23	1.76	<0.0001 *
NO_x_	1.13	0.99	1.28	0.06
NO_2_	1.45	1.28	1.65	<0.0001 *
CO	1.51	1.31	1.74	<0.0001 *
PM_2.5_	0.96	0.86	1.08	0.54
PM_10_	1.22	1.10	1.35	0.0002 *
Decrease vs. stay low				
SO_2_	1.17	1.06	1.29	0.002 *
O_3_	0.96	0.84	1.10	0.53
NO_x_	1.02	0.91	1.15	0.71
NO_2_	1.12	1.01	1.25	0.039 *
CO	0.99	0.89	1.11	0.89
PM_2.5_	0.98	0.87	1.10	0.68
PM_10_	1.11	1.00	1.22	0.043 *

* Significance of *p* < 0.05; Covariates: Preterm birth in first pregnancy; interpregnancy interval; maternal age; race/ethnicity; pre-pregnancy BMI; smoking; alcohol use; parity; insurance status; marital status; asthma history.

**Table 3 ijerph-16-03365-t003:** The adjusted relative risk of preterm birth in second pregnancy by change in whole pregnancy average criteria pollutant exposure levels from first to second pregnancy by prior preterm birth status among 50,005 mothers, NICHD Consecutive Pregnancy Study, 2002–2010.

Change in Whole Pregnancy Exposure from First to Second Observed Pregnancy	Preterm Birth in First Pregnancy				No Preterm Birth in First Pregnancy				Interaction *p*
	RR	LCL	UCL	*p*	RR	LCL	UCL	*p*	
Stay high vs. stay low									
SO_2_	1.21	1.00	1.46	0.052	1.37	1.17	1.59	<0.0001	0.33
O_3_	1.16	0.91	1.47	0.22	1.18	0.96	1.45	0.11	0.90
NO_x_	1.07	0.87	1.30	0.52	1.32	1.12	1.57	0.001	0.10
NO_2_	1.18	0.97	1.43	0.10	1.56	1.34	1.81	<0.0001	0.025 *
CO	1.12	0.92	1.37	0.26	1.41	1.19	1.67	<0.0001	0.08
PM_2.5_	0.93	0.75	1.17	0.54	1.11	0.92	1.35	0.28	0.24
PM_10_	1.10	0.91	1.33	0.32	1.39	1.20	1.60	<0.0001	0.057
Stay moderate vs. stay low									
SO_2_	1.12	0.95	1.32	0.16	1.12	0.99	1.27	0.08	0.97
O_3_	0.95	0.77	1.19	0.68	0.98	0.81	1.17	0.80	0.88
NO_x_	0.93	0.77	1.12	0.44	0.88	0.75	1.03	0.11	0.66
NO_2_	0.98	0.81	1.18	0.81	0.92	0.80	1.06	0.26	0.61
CO	0.93	0.77	1.12	0.45	0.88	0.75	1.03	0.11	0.66
PM_2.5_	0.77	0.64	0.94	0.008	0.85	0.73	1.00	0.05	0.45
PM_10_	1.14	0.97	1.34	0.11	0.99	0.88	1.12	0.93	0.18
Increase vs. stay low									
SO_2_	1.19	1.01	1.40	0.041	1.51	1.33	1.72	<0.0001	0.023 *
O_3_	1.16	0.88	1.53	0.29	1.67	1.33	2.09	<0.0001	0.048 *
NO_x_	1.03	0.85	1.24	0.78	1.19	1.01	1.41	0.042	0.25
NO_2_	1.14	0.93	1.38	0.20	1.64	1.40	1.92	<0.0001	0.004 *
CO	1.15	0.94	1.42	0.18	1.79	1.48	2.16	<0.0001	0.002 *
PM_2.5_	0.86	0.72	1.02	0.07	1.02	0.88	1.19	0.75	0.12
PM_10_	1.21	1.02	1.43	0.028	1.22	1.07	1.40	0.003	0.90
Decrease vs. stay low									
SO_2_	1.03	0.87	1.21	0.73	1.24	1.09	1.40	0.001	0.08
O_3_	0.95	0.78	1.16	0.61	0.96	0.81	1.15	0.67	0.92
NO_x_	0.99	0.83	1.17	0.90	1.04	0.90	1.21	0.59	0.65
NO_2_	1.05	0.88	1.25	0.59	1.15	1.01	1.32	0.04	0.40
CO	0.96	0.81	1.14	0.63	1.01	0.88	1.17	0.86	0.63
PM_2.5_	0.86	0.72	1.02	0.09	1.04	0.90	1.21	0.61	0.10
PM_10_	1.07	0.91	1.26	0.40	1.12	0.99	1.26	0.06	0.67

* Significance of *p* < 0.05. Covariates: Preterm birth in first pregnancy; interpregnancy interval; maternal age; race/ethnicity; pre-pregnancy BMI; smoking; alcohol use; parity; insurance status; marital status; asthma history.

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
