# Peer review of "Air Pollution and Preterm Birth: Do Air Pollution Changes over Time Influence Risk in Consecutive Pregnancies among Low-Risk Women?"

_ijerph, 2019, doi:10.3390/ijerph16183365_

Round 1

Reviewer 1 Report

The work is interesting and shows us the health effect of decreased air pollution. It is good that the authors present the data related to both whole pregnancy exposure and exposure windows to the first 28 weeks of gestation. The results can be acceptable. 

I would suggest that the authors present the detailed description of how to assess the individual exposure level according to community survey on air quality. If the study subjects changed the living or working area during the study period, how the authors calculated their exposure. We know there is generally variation of air quality daily, monthly or yearly, how the authors assess the stable exposure, or varied exposure.

Reviewer 2 Report

I'm really happy to have the possibility to review this paper. This is a quality, well-written article and relevant to public health. I have only one question to the authors, which is written in the PDF.

Reviewer 3 Report

In the present study, the authors propose to evaluate the risk of premature pregnancies as a result of air quality. There is no doubt about the relevance of this type of research. However, I consider that for the manuscript to be published, some details must be adjusted, which I mention below:
1. It must be explained how the relative risk was calculated
2. Given that data were obtained for 8 years, it would be interesting to establish time series with the criteria pollutants, in the same way to assess whether seasonal changes affected the increase in premature pregnancies.
3. Will it be possible to consider meteorological data ?, this with the purpose of estimating some pattern between premature pregnancies and weather changes etc.

Round 2

Reviewer 3 Report

the authors have taken into account the recommendations